# The Characterization and Evaluation of the Soluble Triggering Receptor Expressed on Myeloid Cells-like Transcript-1 in Stable Coronary Artery Disease

**DOI:** 10.3390/ijms241713632

**Published:** 2023-09-04

**Authors:** Zaida Bayrón-Marrero, Siobhan Branfield, Javier Menéndez-Pérez, Benjamín Nieves-López, Laura Ospina, Yadira Cantres-Rosario, Loyda M. Melendez, Robert Hunter, Angelia Gibson, Gerónimo Maldonado-Martínez, A. Valance Washington

**Affiliations:** 1Department of Biology, University of Puerto Rico–Rio Piedras, San Juan, PR 00936, USA; zaida.bayron@upr.edu (Z.B.-M.); sbranfield@oakland.edu (S.B.); jmenendez@oakland.edu (J.M.-P.); benjamin.nieves1@upr.edu (B.N.-L.); laura.ospina@upr.edu (L.O.); 2Department of Biology, Oakland University, Rochester Hills, MI 48309, USA; 3Translational Proteomics Center, Comprehensive Cancer Center, University of Puerto Rico, San Juan, PR 00936, USA; yadira.cantres@upr.edu (Y.C.-R.); loyda.melendez@upr.edu (L.M.M.); 4Department of Microbiology and Medical Zoology, University of Puerto Rico Medical Sciences Campus, San Juan, PR 00936, USA; 5Retroviral Research Center, Universidad Central del Caribe, Bayamón, PR 00960, USA; huntermellado@gmail.com; 6Division of Natural Sciences, Maryville College, Maryville, TN 37804, USA; angelia.gibson@maryvillecollege.edu; 7Exercise Sciences Program, Universidad del Sagrado Corazón, San Juan, PR 00914, USA; geronimo.maldonado@sagrado.edu; 8William Beaumont School of Medicine, Oakland University, Rochester Hills, MI 48309, USA

**Keywords:** triggering receptor expressed in myeloid cells-like transcript (TLT-1), blood platelets, cardiovascular diseases, ventricular function-left, biomarker, anthracyclines

## Abstract

Platelets play crucial roles in the development and progression of coronary artery disease (CAD). The triggering receptor expressed in myeloid cells-like transcript-1 (TLT-1) is stored in platelet α granules, and activated platelets release a soluble fragment (sTLT-1). We set out to better characterize the constituent amino acids of sTLT-1 and to evaluate sTLT-1 for use as a biomarker in patients with stable CAD. We evaluated sTLT-1 release using immunoprecipitation and mass spectrometry and employed statistical methods to retrospectively correlate sTLT-1 concentrations, utilizing ELISA in plasma samples from 1510 patients with documented stable CAD. We identified TLT-1 residues to 133 in platelet releasates. ADAM17 cuts TLT-1, suggesting that S136 is the C-terminal amino acid in sTLT-1. Our results revealed that for CAD patients, sTLT-1 levels did not differ significantly according to primary outcomes of death or major cardiac event; however, patients with left ventricular (LV) dysfunction had significantly lower plasma sTLT-1 levels as compared to those with normal LV function (981.62 ± 1141 pg/mL vs. 1247.48 ± 1589 pg/mL; *p* = 0.003). When patients were stratified based on sTLT-1 peak frequency distribution (544 pg/mL), a significant association with congestive heart failure was identified (OR = 2.94; 1.040–8.282; *p* = 0.042), which could be explained by LV dysfunction.

## 1. Introduction

TLT-1, a type I transmembrane receptor stored in platelet α-granules, localizes rapidly and abundantly to activated platelet membranes. It is one of the most prominent proteins shed from the platelet surface, with an estimated 50,000 surface copies [1,2,3,4,5,6,7]. Our studies have demonstrated that TLT-1 binds fibrinogen, facilitates platelet aggregation, and mediates thrombus formation [2,3].Upon cleavage from the surface, TLT-1 exists in a soluble form (sTLT-1) which is easily detectable in serum (<15 kDa). Although the specific mechanism of TLT-1 cleavage remains unclear, ADAM17 (a disintegrin and metalloprotease 17) has been identified as a potential enzyme responsible [4]. Notably, sTLT-1 is only released from activated platelets, and its measurement in plasma has been utilized as a signature for platelet activation and as an indicator of active participation in the inflammatory process [2,3]. From a pathophysiological perspective, TLT-1, due to its influence on fibrinogen deposition, may also contribute to the development of atherosclerotic lesions considering the well-established role of fibrinogen in atherosclerosis progression [8,9].

Significantly elevated plasma levels of sTLT-1 have been observed in patients presenting with chest pain who were subsequently diagnosed with acute coronary syndrome (ACS) upon entering the emergency room [2]. This difference in sTLT-1 levels alludes to the potential involvement of sTLT-1 in atherothrombosis within ACS and suggests its potential as a biomarker in cardiovascular disease. Other studies have demonstrated associations between sTLT-1 concentrations and cardiovascular disease (CVD) outcomes [10,11]. For instance, Das et al. discovered significant positive associations between sTLT-1 levels and the severity and risk of a coronary event [10].

In order to gain deeper insights into the implications of sTLT-1 in CVD, it is important to understand the molecular constitution of the soluble fragment and whether there are differences in sTLT-1 concentrations between patients with chronic versus acute disease. To this end, we conducted mass spectrometry analysis to characterize the TLT-1 soluble fragment. Angiotensin-converting enzyme inhibitor therapy has a proven clinical benefit for left ventricular dysfunction and congestive heart failure [12,13,14], and we obtained plasma samples from the NIH Biorepository and measured sTLT-1 levels in samples collected from 1510 patients enrolled in the Prevention of Events with Angiotensin-Converting Enzyme inhibitor therapy (PEACE) clinical trial [15]. This trial aimed to compare the effects of an antihypertensive angiotensin-converting enzyme inhibitor (Trandolapril) against placebo in a cohort of 8290 patients with stable coronary heart disease and preserved left ventricular function.

## 2. Results

### 2.1. Defining the TLT-1 Soluble Fragment

Soluble TLT-1 appears as a <15 kDa diffuse band on SDS-PAGE; however, it is not known which residues remain associated with sTLT-1 once cleaved. In order to identify the residues that compose sTLT-1, we used the anti-TLT-1 antibody, AB69 [1], to immunoprecipitate sTLT-1 from the releasate of platelets. In several trials, collagen repeatedly demonstrated the greatest sTLT-1 release; therefore, we activated platelets with collagen. After immunoprecipitation, we evaluated immunoprecipitants by mass spectrometry. Figure 1 demonstrates the four peptides identified in the MS analysis, with the C-terminal residue being lysine 133, which is consistent with trypsin digestion. The cut site residue L137 yields a size consistent with the bands seen on polyacrylamide gels.

### 2.2. Baseline Characteristics of Patient Population

Baseline sTLT-1 concentrations were measured from plasma samples of 1510 patients from the PEACE study, of whom 578 were on Trandolapril and 932 on placebo. Table 1 shows the average baseline TLT-1 plasma concentration according to patient characteristics, treatment history, and study arm (Trandolapril) for the patient samples in our analysis. The mean age of the participants was 64 ± 8 years with a male to female ratio of 82:18. Smokers had lower average plasma sTLT-1 concentrations as compared to nonsmokers (*p* = 0.02), while patients who used beta blockers had significantly higher sTLT-1 levels. There were no other significant differences in baseline sTLT-1 concentrations between treatment groups or other history items.

### 2.3. Association of Patient sTLT-1 Levels with Primary and Secondary Outcomes

Table 2 shows that there were no significant differences between sTLT-1 concentrations according to the study’s primary outcomes of cardiovascular (CV) death (*p* = 0.26) or death from other reasons (death: *p* = 0.39). When sTLT-1 concentrations were compared according to the secondary outcomes listed in Table 2, mean sTLT-1 levels were significantly lower only among the patients with qualitatively abnormal left ventricular function (LVF; 982 ± 1141 pg/mL vs. 1247 ± 1589 pg/mL; *p* = 0.021). Individuals with myocardial infarction trended toward having lower levels (961 ± 1399 pg/mL vs. 1188 ± 1475 pg/mL; *p* = 0.07). There were no direct associations with such secondary outcomes as unstable angina (*p* = 0.26), new onset diabetes (*p* = 0.47), congestive heart failure (*p* = 0.58), or percutaneous coronary intervention (*p* = 0.76).

We proceeded to evaluate our data based on the peak frequency distribution of the PEACE cohort at baseline, setting the cutoff value at 544 pg/mL. There were no significant clinical correlations between the cutoff point of sTLT-1 levels and baseline characteristics (Table 3). Using the 544 pg/mL cutoff to analyze primary and secondary outcomes (Table 4), we found statistically significant *p*-values with qualitative left ventricular function abnormalities (*p* = 0.01) and statistical tendencies with arrythmia (*p* = 0.09) and coronary artery bypass grafting (*p* = 0.07). Death, CV death, and myocardial infarction did not reach statistical significance.

In our Chi square analysis using the 544 pg/mL cutoff point (n = 1510; Table 4), we found a significant association of sTLT-1 levels with qualitative LVF abnormalities (*p* = 0.01), suggesting that patients with plasma sTLT-1 concentrations higher than 544 pg/mL were 1.5 times more likely to develop abnormal LVF than those with lower sTLT-1 plasma concentrations. Upon adjustment of this model using the Akaike information criterion (AIC) and the outcomes listed in Table 5 in our model, we found that levels above 544 pg/mL proved to be a risk factor for qualitative LVF abnormalities (HR = 1.53 [95%CI: 1.15–2.02]; *p* = 0.003 Table 5). High levels of sTLT-1 were also associated with a history of cigarette use (OR = 1.06; *p* = 0.01).

The incorporation of the follow-up visits allowed us to use a Cox regression model to determine the direction of the hazard ratio. We compared individuals with sTLT-1 levels above 544 pg/mL to those with levels below 544 pg/mL. We found that the probability of developing congestive heart failure (CHF) event is almost three times higher when levels are greater than 544 pg/mL (HR = 2.935; *p* = 0.042). Death does not show statistical significance with sTLT-1 (Table 6).

## 3. Discussion

In order to better understand the role of sTLT-1 as a biomarker, we undertook the molecular characterization of the soluble fragment and the evaluation of its presence in the plasma of individuals with chronic inflammatory disease. Having demonstrated that levels of sTLT-1 higher than 1200 pg/mL are associated with poor prognosis in acute respiratory distress syndrome [1], we believe that it is important to cast a wider net so as to understand whether these levels are typical of all diseases or if they are unique to acute diseases, such as ARDS. Fong et al. defined sTLT-1 as the fourth most abundant protein in the platelet sheddome and determined that ADAM17 is responsible for the liberation of sTLT-1 [4]. Our investigations using general and ADAMs-specific inhibitors did not reveal any novel insights into other proteases that cut TLT-1, but they supported previously published work. In the present study, we used mass spectrometry to determine the amino acid composition of sTLT-1 released from activated platelets. Lysine-133 was the C-terminal most amino acid in the peptides generated from a trypsin digestion, suggesting that sTLT-1 release from platelets is the result of proteolysis between amino acids 133 and 160 at the plasma membrane. ADAM17 cuts N-terminal to leucine and valine residues. This implies that either leucine 137 (which predicts a fragment of 12.8 kDa) or leucine 152 (which predicts a fragment of 15.5 kDa) is the sTLT-1 cleavage site. While both sites have the probability to be utilized, based on fragment size, we propose leucine 137 as the primary ADAM17 cut site, making S136 the predicted C-terminal amino acid. This shows that the recombinant sTLT-1, employed for our previous studies (amino acids 20–146), includes the key residues found in the naturally occurring sTLT-1 [2,3].

Moreover, we analyzed a large cohort of patients with CAD and preserved ejection fraction to explore the possibility of plasma sTLT-1 levels correlating with various prognostics associated with CAD. We found that the participants in this study had fairly high levels of sTLT-1 with the peak distribution centered at 544 pg/mL (distribution figure shown in Appendix A). We discovered only two significant differences between groups. We observed that patients with abnormal LVF had significantly lower sTLT-1 plasma concentrations than patients within a normal LVF range. When we set 544 pg/mL as the cutoff concentration based on the peak frequency distribution of the cohort, we saw significantly higher numbers of patients in the >544 pg/mL group with abnormal LVF as compared to the <544 pg/mL group.

There were two other previous studies that evaluated sTLT-1 in relation to CVD. One showed significantly higher serum sTLT-1 concentrations in acute MI and unstable angina as compared to controls [11]. Since serum is largely a byproduct of platelet activation, there are always abundant levels of sTLT-1 in serum, making it difficult to directly compare this study to our research. The other study, conducted by Das et al., used sTLT-1 plasma levels in 117 individuals and found that levels of sTLT-1 were higher in the coronary artery disease and asymptomatic groups as compared to controls [10]. They also observed inverse association between sTLT-1 levels and left ventricular ejection fraction and a positive association with the thrombolytic index of myocardial infarction score, which is an estimated risk for an acute event within 14–30 days. While research completed by Das et al. [10] directly supports the Esponda et al. [2] study, our present efforts target the implication of having higher sTLT-1 levels at baseline in inflammatory states in general.

This study is interesting because the association between sTLT-1 levels and clinical outcomes identifies sTLT-1 as a potential biomarker for recently developed LV dysfunction. For inclusion into this trial, the individual had to have had an ejection fraction >40% as determined by contrast ventriculography, radionuclide ventriculography, or echocardiography completed within 18 months before entering the trial. At randomization they were measured again, and it is the randomization samples (also considered the first visit) samples that are used here. Data about new diagnoses of LV dysfunction are not available in our data nor the PEACE protocol. For this reason, our data about LV dysfunction is about the patients’ conditions during the randomization visit, not the follow-up visits. Remarkably, sTLT-1 demonstrated a better association with LV dysfunction than high-sensitivity C-reactive protein, one of the most commonly used biomarkers for CAD. That may be because all of these patients were already classified with CAD. Although a documented myocardial infarction at least three months prior to randomization was one of the inclusion criteria, the possibility remains that some of these patients had a silent myocardial infarction. We are unable to discern why these patients develop LV dysfunction, since we did not have access to the data to verify this point.

The Akaike information criterion model suggests that the association of CHF with sTLT-1 can be attributed to LVF abnormalities. This is consistent with the hypothesis of the original PEACE study that patients with normal or slightly reduced left ejection fraction would derive benefit from having an ACE inhibitor added to their normal regiment. Even though all patients with congestive heart failure develop left ventricular dysfunction [16], it is an issue of magnitude of the dysfunction. Patients with a clinical diagnosis of congestive heart failure are managed with medications and know the diagnosis, which is also documented in the medical chart. On the other hand, patients with left ventricular dysfunction are usually detected via a 2-D echocardiography imaging test (2-D echo) [17]. A 2-D echo is conducted ambulatory, and it is strictly a measurement of the contractility of the heart. The dysfunction may be mild, moderate, or severe as well as acute, reversible, or chronic in nature, leaving many patients unaware of their condition. Thus, for the first group, a clinical diagnosis is made, while for the second group, a diagnosis is the result of a test where the finding may be reversible or progressive; acute or chronic; mild or severe. This raises the question of why patients with congestive heart failure do not demonstrate significant differences in sTLT-1 levels between groups. There are several possible explanations, including the use of such medications as aspirin or anticoagulants. Another possibility is that patients with congestive heart failure represent just a fraction of those in the abnormal LVF group, and therefore, there may not be enough subjects to detect a meaningful difference. Finally, a more important possibility is that for subjects with ventricular dysfunction, it may be an acute or subacute process rather than the chronic condition, as it is for the congestive failure group. Thus, the sTLT-1 findings may be related to acute or subacute ventricular dysfunction with platelet activation precipitating thrombocytopenia, while congestive heart failure may be chronic and relatively stable. As such, in individuals with abnormal LVF and plasma sTLT-1 levels above 544 pg/mL, the risk of developing congestive heart failure may actually be greater than the three-fold chance predicted in this study.

This study adds understanding to our understanding of the meaning of sTLT-1 found in plasma. Not only in CVD, but it provides a comparison for other diseases where sTLT-1 is measured as potential platelet-based biomarker such as acute respiratory syndrome.

## 4. Materials and Methods

All studies using human subjects were approved by the University of Puerto Rico Institutional Review Board based on the Declaration of Helsinki.

### 4.1. Patient Population

We quantified sTLT-1 in plasma concentrations of individuals between 50 and 83 years of age with documented stable cardiovascular disease. They were enrolled in the PEACE clinical trial with a median follow-up of 4.8 years. The selection criteria for the population included a history of coronary artery disease and percutaneous transluminal coronary angioplasty documented by at least one of the following: MI, coronary-artery bypass grafting, PCI at least 3 months before enrollment, or left ventricular ejection fraction >40%. Details of the study can be found in Reference [15].

### 4.2. Biomarker Analyses

We evaluated 3549 samples from the PEACE study; 320 were lost due to computer malfunction at the point of data entry, of which 240 were first-visit samples and 80 were follow-up visits. In the end, we reported 1510 first-visit samples and 1719 follow-up samples. We considered the first visit after randomization as baseline (6 ± 1 months) [17]. Plasma was frozen and shipped to a central laboratory, where samples were stored at −80 °C. Aliquots were shipped frozen on dry ice to the University of Puerto Rico, where the sTLT-1 was analyzed as in Reference [1]. Soluble TLT-1 was analyzed using the Bio-techne Human TREML1/TLT-1 DuoSet ELISA kit (Cat# DY2394; assay range 15.6–1000 pg/mL) according to manufacturer’s protocol. Briefly, in a 96-well plate coated with the first monoclonal antibody (goat anti-human TREML1) overnight then blocked with 1% bovine serum albumin. Next, 20 μL of plasma were added to 96-well plates, a volume of 100 μL of standard TLT-1 and a serial dilution was made for reference. The plates were incubated for 2 h at room temperature and were then washed with phosphate-buffered saline (PBS). We then added 100 μL of solution of the second monoclonal antibody (biotinylated goat anti-human TREML1) and incubated it for 2 h at room temperature and washed with PBS. Then 100 μL of streptavidin-horseradish peroxidase was added to each well and incubated for 20 min at room temperature. After PBS washing, 100 μL of tetramethylbenzidine was added to each well and the mixture was left for 20 min at room temperature. The enzyme reaction was stopped by the addition of 50 μL of 1N H2SO4, and absorbance at 540 nm was measured on a Tecan Infinite^®^ 200 PRO. The intra assay coefficient of variation was 14.85/35 plates, and the inter-assay coefficient of variation average was 2.6/plate. Results were reported in pg/mL.

### 4.3. Platelet Preparation

Platelets were prepared as previously described in Reference [3].

### 4.4. Mass Spectrometry Analysis

LC-MS/MS Analysis: Easy-nLC1200 (Thermo Fisher Scientific, Waltham, MA, USA) with a PicoChip H354 REPROSIL-Pur C18-AQ 3 μM 120 A (75 μm × 105 mm) chromatographic column was used for peptide separation. The separation was obtained using a gradient of 5–40% of 0.1% of formic acid in 80% acetonitrile using a flow rate of 300 nL/min with a maximum pressure of 300 bar and the injection volume of 2 μL per sample. MS/MS analysis utilized Q-Exactive Plus (Thermo Fisher Scientific) operated in positive polarity mode and a data-dependent mode. The full scan/MS1 was measured over the range of 400 to 1600 *m*/*z*. The MS2 (MS/MS) analysis was configured to select the 10 most intense ions for fragmentation. A dynamic exclusion parameter was set for 15 s. Database Search: Once the mass spectrometry analysis was finished, the samples were searched with a Homo sapiens database downloaded from UniProt (Universal Protein Resource) on 9 February 2021. The raw data were analyzed using Proteome Discoverer software version 2.5. A dynamic modification for oxidation +15.995 Da (M) and a static modification of +57.021 Da (C) generated by the alkylation during processing was included in the parameters for the search.

### 4.5. Statistical Analysis

A normality diagnostic for all variables was completed using the Shapiro–Wilk estimator. Frequencies, percentages, central tendency, and dispersion measures were calculated to assess the raw distribution of the variables. In order to evaluate mean ranks differences in TLT-1 levels, a Mann–Whitney test was performed. For the multivariate scheme, a Cox regression model using hazard ratios was designed to compare levels of sTLT-1 against comorbidities. The binary scenario for this factor was setting sTLT-1 levels above 544 pg/mL as one and levels below 544 pg/mL as zero; 544 pg/mL was chosen based on the peak frequency distribution with a bin set at 5 pg/mL. Model fitness was assessed using the Akaike information criterion (AIC). All tests were performed using SPSS version 28.0, R Studio version 4.0.4. and GraphPad Prism version 9.1.0. Statistical significance was set to a *p* ≤ 0.05, except for the normality diagnostics.

## 5. Conclusions

We propose that sTLT-1 consists of extra cellular domain of sTLT-1 to amino acid 137 and that elevated sTLT-1 levels in the evaluated cohort of patients suggest that sTLT-1 may have utility as a biomarker for CHF prognosis. There is a growing concern over cardiac damage associated with anthracyclines and trastuzumab treatment in cancer patients. Since studies have shown that a dose-dependent cardiac damage and death are associated with LV dysfunction and heart failure [18,19,20,21], it seems prudent to analyze sTLT-1 levels in the same cohort of patients to assess whether changes in sTLT-1 level are predictive in this clinical setting.

## Figures and Tables

**Figure 1 ijms-24-13632-f001:**
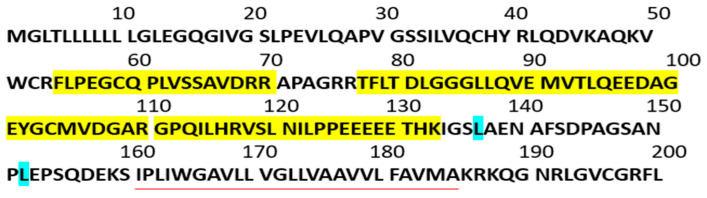
Characterization of the TLT-1 soluble fragment (sTLT-1). The sequence of the TLT-1 extracellular domain (amino acids [aa] 20–160) showing peptides identified by mass spectrometry is highlighted in yellow. Based on these identified peptides, we show the potential ADAM17 leucine cleavage sites, highlighted in blue, and predict L137 as the potential ADAM17 primary cleavage site. The red line represents the transmembrane domain.

**Table 1 ijms-24-13632-t001:** Baseline characteristics of the PEACE study patient population.

Baseline Characteristic		sTLT-1 Average (pg/mL)	*p*-Value
Cigarette Use	YES NO	(N = 1156) 1140 ± 1472 (N = 353) 1290 ± 1454	**0.019**
History of myocardial infarction	YES NO	(N = 903) 1130 ± 1395 (N = 607) 1244 ± 1570	0.08
History of Diabetes	YES NO	(N = 231) 1298 ± 1613 (N = 1279) 1153 ± 1441	0.098
History of CABG	YES NO	(N = 556) 1180 ± 1515 (N = 954) 1173 ± 1442	0.46
History Of PCI	YES NO	(N = 693) 1093 ± 1316 (N = 817) 1245 ± 1584	0.09
History of Stroke	YES NO	(N = 60) 821 ± 730 (N = 1450) 1190 ± 1490	0.12
Tandolapril	YES NO	(N = 733) 1214 ± 1469 (N = 777) 1139 ± 1400	0.77
History of Angina	YES NO	(N = 1091) 1183 ± 1532 (N = 419) 1157 ± 1290	0.57
History of Hbp	YES NO	(N= 618) 1051 ± 1220 (N = 892) 1262 ± 1614	0.13
Antiplatelet use	YES NO	(N = 701) 1338 ± 1617 (N = 44) 1390 ± 1320	0.75
Anticoagulant use	YES NO	(N = 28) 1311 ± 1601 (N = 432) 1353 ± 1801	0.94
Anticoagulant and antiplatelet use	YES NO	(N = 21) 1578 ± 1773 (N = 439) 1340 ± 1789	0.25
Beta blockers use	YES NO	(N = 222) 1507 ± 2060 (N = 156) 1130 ± 1462	**0.019**
Serum cholesterol	<200 mg/dL >200 mg/dL	(N = 376) 1379 ± 1570 (N = 230) 1209 ± 1467	0.07

Student’s *t* test comparing the means for the characteristics of the 1510 patients from the first follow-up visit used in our study. “YES” denotes those with the baseline characteristic, while “NO”—those without them. “N” for each group is listed in parentheses. *p*-values ≤ 0.05 are considered significant and are written in red.

**Table 2 ijms-24-13632-t002:** AveragesTLT-1 levels according to primary and secondary outcomes from the PEACE study.

Outcomes		sTLT-1 Average (pg/mL)	*p*-Value
**Primary outcomes**			
Death	YES NO	(N = 66) 1014 ± 1105 (N = 1444) 1183 ± 1484	0.16
Cardiovascular death	YES NO	(N = 35) 982 ± 1189 (N = 1475) 1180 ± 1475	0.26
Other death	YES NO	(N = 31) 1050 ± 1020 (N = 1479) 1179 ± 1477	0.39
**Secondary outcomes**			
Myocardial infarction	YES NO	(N = 79) 961 ± 1002 (N = 1431) 1188 ± 1490	0.07
New diabetes	YES NO	(N = 117) 1143 ± 1336 (N = 1393) 1179 ± 1480	0.47
Congestve heart failure	YES NO	(N = 32) 1191 ± 1719 (N = 1478) 1176 ± 1464	0.58
Coronary artery bypass grafting	YES NO	(N = 107) 1273 ± 1399 (N = 1403) 1169 ± 1475	0.98
Unstable angina	YES NO	(N = 188) 1120 ± 1281 (N = 1322) 1184 ± 1494	0.26
Arrythmia	YES NO	(N = 61) 1293 ± 1533 (N = 1449) 1171 ± 1467	0.96
Percutaneous coronary intervention	YES NO	(N = 182) 1138 ± 1464 (N = 1328) 1181 ± 1470	0.76
Left ventricle qualitative abnormal function	YES NO	(N = 244) 982 ± 1141 (N = 1022) 1247 ± 1589	**0.021**

Table shows the average sTLT-1 levels with the standard deviation. Student t-test was used to determine the significance of the association. “YES” denotes the number of patients with the listed outcome, while “NO” denotes those without them. “N” for each group is listed in parentheses. *p*-values ≤ 0.05 are considered significant and are written in red.

**Table 3 ijms-24-13632-t003:** Baseline characteristics of the PEACE study patient cohort using peak frequency distribution of 544 pg/mL to stratify patient population.

Baseline Characteristic	sTLT-1 < 544 pg/mL	sTLT-1 ≥ 544 pg/mL	*p*-Value
N (1510)	578 (38%)	932 (61%)	
AGE (1510)	63 ± 7 (578)	63.99 ± 8 (932)	0.97
Female sex (260)	96	164	0.62
Body mass index (1510)	28 ± 4 (575)	28.89 ± 4 (926)	0.72
History of angina (1091)	418	673	0.96
History of myocardial infarction (903)	360	543	0.12
Total cholesterol (1470)	192 ± 38 (563)	190 ± 37 (907)	0.35
Cigarette use (1156)	454	702	0.16
History of diabetes (231)	77	154	0.09
History of blood pressure (618)	240	378	0.71
History of coronary artery bypass grafting (556)	223	333	0.26
History of percutaneous coronary intervention (693)	278	415	0.18
History of stroke (60)	25	35	0.58
Tandolapril (733)	275	458	0.56

Chi square was used to determine the significance of the association. *p*-values ≤ 0.05 are considered significant.

**Table 4 ijms-24-13632-t004:** Multivariate analysis evaluating correlation of sTLT-1 with primary and secondary outcomes.

Outcomes	sTLT-1 < 544 pg/mL	sTLT-1 ≥ 544 pg/mL	*p*-Value
**Primary outcomes**			
Death	22	44	0.40
Cvdeath	13	22	0.89
**Secondary outcomes**			
Unstable angina	67	121	0.43
Arrythmia	17	44	0.09
Myocardial infarction	31	48	0.86
Left ventricle qualitatively abnormal function	113	131	**0.01**
New diabetes mielitus	47	70	0.66
Coronary artery bypass grafting	32	75	0.07
Percutaneous coronary intervention	68	114	0.79
Congestive heart failure	14	18	0.52
Acs (mi, ptca, cabg, stroke, cvdeath, ua)	136	249	0.17

Chi square analysis of unadjusted risks of mortality with primary and secondary outcomes. *p*- values ≤ 0.05 are considered significant and are written in red.

**Table 5 ijms-24-13632-t005:** Adjusted Correlations with sTLT-1 and secondary outcomes.

Risk Factors for Cardiovascular Diseases (1510)		Myocardial Infarction	Unstable Angina	Left Ventricle Qualitative Abnormal Function	Percutaneous Coronary Intervention	Coronary Artery Bypass Grafting	Congestive Heart Failure	Death	Cardiovascular Death	Acute Coronary Syndrome
**sTLT ≥ 544 pg/mL**	Or (95% CI) *p*-value	1.04 (0.66–1.66) 0.857	0.88 (0.64–1.21) 0.426	**1.53 (1.15–2.02) 0.003**	0.96 (0.70–1.32) 0.786	0.67 (0.44–1.03) 0.066	1.26 (0.62–2.55) 0.521	0.80 (0.47–1.35) 0.399	0.95 (0.48–1.90) 0.889	0.84 (0.66–1.07) 0.167

The adjustment employs the Akaike information criterion (AIC) model with using the secondary outcomes of MI, LV abnormal function, percutaneous coronary intervention, arrythmia, unstable angina, congestive heart failure and cigarette smoking. *p*-values ≤ 0.05 are considered significant and are written in red. The values are reported as odds ratio (95% confidence interval [Cl]).

**Table 6 ijms-24-13632-t006:** Predictive validation of plasma sTLT-1 levels associated with primary and secondary outcomes using 544 pg/mL as a cutoff.

Outcomes	Stlt-1 < 544 pg/mL	sTLT-1 ≥ 544 pg/mL	*p*-Values
Myocardial infarction	1 (reference)	0.695 (0.413–1.170)	0.171
Unstable angina	1 (reference)	0.842 (0.603–1.175)	0.312
Stroke	1 (reference)	4.214 (0.543–32.691)	0.169
New diabetes mellitus	1 (reference)	1.235 (0.803–1.900)	0.336
Cv death	1 (reference)	1.737 (0.800–3.773)	0.163
Percutaneous coronary intervention	1 (reference)	1.045 (0.750–1.455)	0.796
Coronary artery bypass grafting	1 (reference)	0.652 (0.416–1.021)	0.062
Congestive heart failure	1 (reference)	2.935 (1.040–8.282)	**0.042**
Arrhythmia	1 (reference)	0.784 (0.388–1.585)	0.498

Cox regression curves comparing probabilities of developing specific cardiovascular outcomes with higher levels of s-TLT-1 (n = 1510). The values are hazard ratios (95% confidence interval (Cl)). The outcomes listed in the table are the variables included in our model. *p*-values ≤ 0.05 are considered significant and written in red.

## Data Availability

Not applicable.

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
