# Peer review of "The Characterization and Evaluation of the Soluble Triggering Receptor Expressed on Myeloid Cells-like Transcript-1 in Stable Coronary Artery Disease"

_ijms, 2023, doi:10.3390/ijms241713632_

Round 1
Reviewer 1 Report (Previous Reviewer 2)
all the questions have been answered.
Author Response
We thank the reviewer for critiquiing our manuscript
Reviewer 2 Report (New Reviewer)
The authors investigated plasma sTLT-1 levels in 1510 patients with CAD. However, I have some comments.
1) The only significant finding in this study is that patients with LV dysfunction had significantly lower plasma sTLT-1 levels. First, please describe the definition of LV dysfunction, one of the seconary outcomes. Did such patients develop LV dysfunction during the follow-up period? Was there any significant correlation between sTLT-1 levels and LV ejection fraction?
2) Why did some patients develop LV dysfunction? Silent myocardial infarction may have occurred to such patients?
3) Was there any correlation between high-sensitivity C-reactive protein, the most commonly used biomarker for CAD, and sTLT-1 levels? Was sTLT-1 level a better predictor than hsCRP levels?
4) Regarding the measurement of sTLT-1 levels, please describe the detection range and the intra-assay and inter-assay coefficients of variation.
5) All parts of this paper seem to be too long and redundant. It can be written much more concisely.
Author Response
We thank the reviewer for taking the time to critique our manuscript. Below we provide a point-by-point response to the reviewer’s critique.
1) First, please describe the definition of LV dysfunction, one of the secondary outcomes.
As taken from the PEACE data dictionary: Ejection fraction < 40% as determined by contrast ventriculography, radionuclide ventriculography, or echocardiography
Did such patients develop LV dysfunction during the follow-up period?
No, it would have been within the 18 months prior to randomization. For inclusion into this trial, the individual had to have had one of the tests listed above completed within 18 months before entering the trial. At randomization they were measured again and these results were from randomization.
Was there any significant correlation between sTLT-1 levels and LV ejection fraction?
There was not.
2) Why did some patients develop LV dysfunction?
This is not addressed in the data provided
Silent myocardial infarction may have occurred to such patients? This is possible
It is possible, but we do not have the access to the data to verify this point. A documented myocardial infarction at least 3 months prior to randomization was one of the inclusion criteria
3) Was there any correlation between high-sensitivity C-reactive protein, the most commonly used biomarker for CAD, and sTLT-1 levels? Was sTLT-1 level a better predictor than hsCRP levels?
There was no correlation between sTLT-1 levels and CRP. Yes, in these patients sTLT-1 was a better predictor than hsCRP.
4) Regarding the measurement of sTLT-1 levels, please describe the detection range and the intra-assay and inter-assay coefficients of variation.
Assay range - 15.6 - 1,000 pg/mL
Intra-assay COV – 14.85 for 35 plates
Inter-assay COV – avg 2.6/plate
5) All parts of this paper seem to be too long and redundant. It can be written much more concisely.
We have reduced the verbiage.
Round 2
Reviewer 2 Report (New Reviewer)
The only significant finding in this study is that patients with LV dysfunction had significantly lower plasma sTLT-1 levels. However, I found no improvement. The authors should describe the answers to my comments in the revised paper.
Author Response
We appreciate your input, and we have integrated your suggestions into the manuscript. All the changes/additions are highlighted in blue. We believe this has significantly enhanced the message. Thank you.
Round 3
Reviewer 2 Report (New Reviewer)
I have no further comments.
This manuscript is a resubmission of an earlier submission. The following is a list of the peer review reports and author responses from that submission.
Round 1
Reviewer 1 Report
This study analyze the possible correlation between level the soluble form of the triggering receptor expressed in myeloid cells like-transcript-1 (sTLT-1), that is secreted by activated platelets, and progression of coronary artery disease, to evaluate whether sTLT-1 could be a biomarker of CAD progression.
Authors found that the soluble form of the triggering receptor expressed in myeloid cells like-transcript-1 (sTLT-1) stored in platelet granules is not correlated with the progression of coronary artery disease. They only found the patients with left ventricular dysfunction had significant lower plasma sTLT-1.
On the bases of these results, authors conclude that s-TLT-1 could be biomarker for heart failure prognosis. Although results may be interesting, my main concern is that authors do not describe other biomarkers, or other biochemical parameters measured in subjects. First of all a marker of platelet activation, P-selectin and /or fibrinogen receptor (GPIIb/IIIa), must be measured; Furthermore, it would be more interesting if other biochemical parameters will be evaluated (such as cholesterol, C reactive protein, electrolyte blood concentration etc. ).
Author Response
Thank you for your comments. We whole heartly agree with your comment. In our lab we spend an enormous amount of time trying to understand the relationship between TLT-1, GPIIb/IIIa and P-selectin to the point that we have double null mice of TLT-1 and GPIIb and TLT-1 and P-selectin and these mice in disease models. This study was derived from our recent study on acute respiratory distress syndrome where the reviewers wanted to know what other levels of sTLT-1 are found in individual with other diseases such as CVD. Specifically, to answer that question we were able to query the NIH biorepository for sample. Unfortunately, we retrospectively received only a small amount of plasma as sample. We did not have enough to measure soluble P-selectin or GPIIb/IIIa.
Reviewer 2 Report
The article conducted a retrospective study on a large cohort and identified a potential biomarker for CHF prognosis, which is of clinical significance. However, further research is lacking on the mechanism of changes in this biomarker.
1) The sentence “From a pathophysiological point of view ……” on line 51-52, any supported literature? Or is there a detailed explanation?
2) Is it possible to reorganize the structure of the entire section of “Introduction” to make the content more coherent and logical?
3) How was the cutoff value of 544 selected?
4) Line 207: any data or supported literature for “lower platelet counts than those who do not suffer from reduced LVF”?
5) Where is Table 7 mentioned on Line 210?
6) Line 247 mentioned a study that used serum to detect, which is more suitable? Serum or plasma?
Minor editing of English language required
Author Response
The article conducted a retrospective study on a large cohort and identified a potential biomarker for CHF prognosis, which is of clinical significance. However, further research is lacking on the mechanism of changes in this biomarker.
1) The sentence “From a pathophysiological point of view ……” on line 51-52, any supported literature? Or is there a detailed explanation?
TLT-1 binds fibrinogen and fibrinogen plays an established role in atherosclerosis progression. We have modified the sentence and added references
2) Is it possible to reorganize the structure of the entire section of “Introduction” to make the content more coherent and logical?
We have re written the introduction based on your request.
3) How was the cutoff value of 544 selected?
It was selected on the peak distribution of patient sTLT-1 levels.
4) Line 207: any data or supported literature for “lower platelet counts than those who do not suffer from reduced LVF”?
Yes, we have added that reference.
5) Where is Table 7 mentioned on Line 210?
We removed a table that had redundant information, and Table 7 became Table 6; we have corrected this error.
6) Line 247 mentioned a study that used serum to detect, which is more suitable? Serum or plasma?
You should not use Serum to measure sTLT-1 levels, sTLT-1 like serum is a by product of platelet activation and because of that there is A LOT of sTLT-1 in serum to the point that we use it as one of our positive controls in our ELISAS.
Reviewer 3 Report
An éloquent study, progressing previously published work. The manuscript is very well written with a solid hypothesis supported by the literature and previous research. The dat is well interrogated, interpreted and presented. The study is further supported by a comprehensive cohort. The findings herein will be of significant interest to a wide readership, and specifically cardiovascular/platelet scientists and clinicians.
Only minor edits to make in that the ELISA methodology should be comprehensively detailed in this paper. It is important for groups working in this area to accurately, specifically and sensitively apply these findings to future studies and research- from basic to translational to clinical.
Author Response
Thank you for your kind remarks and for pointing this oversight out. We have updated our methods.
Reviewer 4 Report
The manuscript reports the evaluation of TLT-1 in stable coronary artery disease. The aim of the study is interesting, however, the paper in its present form, in my opinion, cannot be recommended for publication – major revision is required.
The aim of the study should be more clearly described in the Introduction and in the Abstract. Also, the conclusions from this study should be clearly formulated and relate to the study results.
The methods for sTLT-1 measurements in plasma should be described in detail in M&M. In this study, two methods for TLT-1 evaluation were used, one in platelet lysates when TLT-1 was immunoprecipitated with AB69 antibodies and the next in TLT-1 the peptide identification was analyzed in MS. However, for the plasma level of sTLT-1 the ELISA method was used (the information from Abstract). This was a commercial ELISA test? It is not clear how sTLT-1 was measured in the patient’s plasma. How many patients participated in this study? 1510 or 3229? It is not clear.
In my opinion, the statistical analysis needs improvement. In the Statistical analysis paragraph, the Authors described that an independent t-test was used to evaluate mean differences in TLT-1 level. The normal distribution of data is a condition for using this test. Was this condition met? The wide scatter of results and the consequentially high standard deviations were observed for TLT-1 (for example in Table 1). In the Results, the section was no data for the Kaplan-Meier analysis, but this test is included in the Statistical Analysis paragraph. Additionally, the names of statistical tests should be worded in legends to tables.
The Authors interchangeably used the words “correlation” and “association”, but for the description of results should be used words “differences between group 1 and 2 in TLT-1 level” or the association between the two categorical variables (Chi2 test).
When the OR or HR values are described also the confidence interval (CI) should be included (pp. 5, 6).
In the note to Table 2 is Kruskal-Wallis test. Why?
How the TLT-1 544 pg/ml cutoff was calculated? The figure showing the sTLT-1 distribution should be included in the manuscript.
Table 4 shows the same data as Table 3.
Table 1. and Table 2: the note “+/-“ should be replaced with a symbol “±”. Also, when the value is high (1507) the decimal places are unnecessary.
pp. 3, line 112, and pp. 4 line 114 units are lacking.
Author Response
The aim of the study should be more clearly described in the Introduction and in the Abstract. Also, the conclusions from this study should be clearly formulated and relate to the study results.
We have rewritten this to make it clearer.
The methods for sTLT-1 measurements in plasma should be described in detail in M&M. In this study, two methods for TLT-1 evaluation were used, one in platelet lysates when TLT-1 was immunoprecipitated with AB69 antibodies and the next in TLT-1 the peptide identification was analyzed in MS. However, for the plasma level of sTLT-1 the ELISA method was used (the information from Abstract). This was a commercial ELISA test? It is not clear how sTLT-1 was measured in the patient’s plasma. How many patients participated in this study? 1510 or 3229? It is not clear.
Thank you for point this out. We updated our methods with the name and details of the ELISA. There were 1510 patients and with repeat longitudinal samples the total number was 3229. We have added clarification in the text.
In my opinion, the statistical analysis needs improvement. In the Statistical analysis paragraph, the Authors described that an independent t-test was used to evaluate mean differences in TLT-1 level. The normal distribution of data is a condition for using this test. Was this condition met? The wide scatter of results and the consequentially high standard deviations were observed for TLT-1 (for example in Table 1). In the Results, the section was no data for the Kaplan-Meier analysis, but this test is included in the Statistical Analysis paragraph. Additionally, the names of statistical tests should be worded in legends to tables.
Yes, to evaluate mean ranks differences in TLT-1 levels, a Mann-Whitney test was performed. We have updated the methods and removed the Kaplan Meier analysis section in the statistical analysis. We have added the statistical tests in the table legends.
The Authors interchangeably used the words “correlation” and “association”, but for the description of results should be used words “differences between group 1 and 2 in TLT-1 level” or the association between the two categorical variables (Chi2 test).
Thank you. We have corrected this language throughout the text.
When the OR or HR values are described also the confidence interval (CI) should be included (pp. 5, 6).
This has been corrected. Thank you.
In the note to Table 2 is Kruskal-Wallis test. Why?
We used this because Kruskal Wallis test because is a non-parametric test.
How the TLT-1 544 pg/ml cutoff was calculated? The figure showing the sTLT-1 distribution should be included in the manuscript.
The distribution was calculated using Prism and confirmed with excel and a bin size of 5 pg/mL. This has been added to the materials and methods. The figure has been added to supplemental data.
Table 4 shows the same data as Table 3.
This has been fixed.
Table 1. and Table 2: the note “+/-“ should be replaced with a symbol “±”. Also, when the value is high (1507) the decimal places are unnecessary.
This has been fixed.
- 3, line 112, and pp. 4 line 114 units are lacking.
This has been fixed.
Round 2
Reviewer 1 Report
Authors did not address my question, experiments demonstating platelet activation , through Pselectin and/or fibrinogen receptor expression are crucial to support conclusion.
May be minor revision
Author Response
While we agree the addition of those parameters would add to the complexity to the study, we are unable to do these assays. The plasma that we received was frozen and all platelets were removed. The fact that sTLT-1 is in the plasma demonstrates that the platelets were activated, since platelets do not have TLT-1 on the surface if they are resting. In fact, it has been shown that TLT-1 is a better marker of activation than P-selectin (DOI 10.1182/bloodadvances.2018017756). The fibrinogen receptor, the integrin GP2b3a, is constitutively found on the surface of the platelets, so unless the patients have Glanzmann's Thrombasthenia, the fibrinogen receptor is present.
Moreover, sTLT-1 was measured retrospectively. We were given an aliquot of the plasma to measure sTLT-1 by ELISA and the data from the original study, so we could only do the ELISA and compare our findings with the already published findings. We were UNABLE to measure either P-selectin or fibrinogen receptor expression in this cohort.
There are many papers that are published looking at only one platelet marker, P-selectin or the fibrinogen receptor are not necessary for the interpretation of our results.
https://www.nmcd-journal.com/article/S0939-4753(19)30281-9/fulltext#secsectitle0055
https://journals.plos.org/plosone/article?id=10.1371/journal.pone.0141693
https://pubmed.ncbi.nlm.nih.gov/25301077/
https://www.ncbi.nlm.nih.gov/pmc/articles/PMC7894689/
Reviewer 4 Report
Table 2. I don't understand why Kruskal-Wallis test was used. The Kruskal-Wallis test allows for comparing 3 or more groups (this is non-parametric ANOVA). In my opinion for the data presented in Table 2, the t-independent test or Mann-Whitney test should be used.
Author Response
As per your request, we have used the t-test in place of the Kruskal-Wallis ANOVA test for table - 2